# DOG: Discriminator-only Generation Beats GANs on Graphs

## Abstract

We propose discriminator-only generation (DOG) as a generative modeling approach that bridges the gap between energy-based models (EBMs) and generative adversarial networks (GANs). DOG generates samples through iterative gradient descent on a discriminator's input, eliminating the need for a separate generator model. This simplification obviates the extensive tuning of generator architectures required by GANs. In the graph domain, where GANs have lagged behind diffusion approaches in generation quality, DOG demonstrates significant improvements over GANs using the same discriminator architectures. Surprisingly, despite its computationally intensive iterative generation, DOG produces higher-quality samples than GANs on the QM9 molecule dataset in less training time.

## 1 Introduction

Generative modeling approaches, such as denoising diffusion (Sohl-Dickstein et al., 2015; Ho et al., 2020) and GANs (Goodfellow et al., 2014; Sauer et al., 2023), have revolutionized content creation across various domains. Given their wide-ranging potential, it is essential to thoroughly explore their design space.

GANs consist of a generator, which generates samples from noise vectors to match a real data distribution, and a discriminator, which assigns a realness score to distinguish between real and generated samples. Notably, the discriminator is discarded after training, leaving only the generator in use. Optimizing generator architectures requires substantial effort (Karras et al., 2019; 2020; 2021; Walton et al., 2022; De Cao & Kipf, 2018; Krawczuk et al., 2021), resulting in complex settings with multiple generators being used in conjunction in recent graph GANs (Martinkus et al., 2022). We explore DOG, an approach that removes the need for a generator altogether and instead directly leverages the information stored in a discriminator, aligning it conceptually with EBMs (Ackley et al., 1985; Xie et al., 2016; 2017; Du & Mordatch, 2019), and with refining generated samples using a discriminator (Tanaka, 2019).

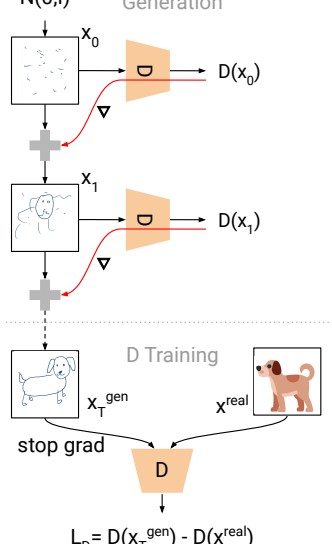

Figure 1: For generation, DOG optimizes a sample directly through gradient descent w.r.t. a generation loss on the output of the discriminator $D$. Training involves alternating between sample generation (top) and discriminator updates (bottom), akin to a GAN, but without the need for a generator model.

DOG uses only a single discriminator model for generation. It starts with a pure noise sample in the target domain and optimizes it directly by gradient descent w.r.t. a generation loss on the output of the discriminator. The generated samples, along with real samples, are then used to train the discriminator by alternating sample generation and discriminator updates, as shown in Figure 1. In contrast to GANs, where a generator and a discriminator act as adversaries, the DOG discriminator serves as its own adversary.

Our primary contribution is demonstrating that DOG outperforms GANs in graph generation without necessitating adjustments to the discriminator architecture or the original GAN hyperparameters. For deeper insight, we conduct a convergence analysis of DOG and illustrate it on a 2D toy dataset.

---

**Algorithm 1** Pseudocode for DOG with Wasserstein loss and a batch size of 1. For a PyTorch version, see Algorithm 2 (Appendix).

---

1: **function** $\text{GO}(D_{\boldsymbol{\theta}}, \mathbf{x}_0^{\text{gen}})$            ▷ Generation optimization
2:     **for** $t \in [0, 1, \ldots, T-1]$ **do**
3:        $L_G \leftarrow -D_{\boldsymbol{\theta}}(\mathbf{x}_t^{\text{gen}})$
4:        $\mathbf{x}_{t+1}^{\text{gen}} \leftarrow \text{optimizer}_{\mathbf{x}}(\mathbf{x}_t^{\text{gen}}, \nabla_{\mathbf{x}_t^{\text{gen}}} L_G)$          ▷ E.g., Equation (2)
5:     **end for**
6:     **return** $\mathbf{x}_T^{\text{gen}}$
7: **end function**

8: Randomly initialize discriminator parameters $\boldsymbol{\theta}$          ▷ Training
9: **while** training not done **do**
10:     $\mathbf{x}^{\text{real}} \sim$ training set
11:     $\mathbf{x}^{\text{gen}} \leftarrow \text{GO}(D_{\boldsymbol{\theta}}, \boldsymbol{\epsilon})$ with $\boldsymbol{\epsilon} \sim \mathcal{P}$
12:     $L_D \leftarrow D_{\boldsymbol{\theta}}(\text{sg}(\mathbf{x}^{\text{gen}})) - D_{\boldsymbol{\theta}}(\mathbf{x}^{\text{real}})$          ▷ sg: stop gradient
13:     $\boldsymbol{\theta} \leftarrow \text{optimizer}_D(\boldsymbol{\theta}, \nabla_{\boldsymbol{\theta}} L_D)$
14: **end while**

15: $\mathbf{x}^{\text{gen}} \leftarrow \text{GO}(D_{\boldsymbol{\theta}}, \boldsymbol{\epsilon})$ with $\boldsymbol{\epsilon} \sim \mathcal{P}$          ▷ Inference

---

## 2   METHOD: DISCRIMINATOR-ONLY GENERATION

GANs rely on two loss functions: $L_G$, which incentivizes the generator $G$ to generate samples $\mathbf{x}^{\text{gen}}$ that receive high scores from the discriminator $D$, and $L_D$, which encourages $D$ to assign low scores to $\mathbf{x}^{\text{gen}}$ and high scores to real samples $\mathbf{x}^{\text{real}}$. An example is the Wasserstein loss (Arjovsky et al., 2017):

$$L_G = -\mathbb{E}[D(\mathbf{x}^{\text{gen}})] \quad \text{and} \quad L_D = \mathbb{E}[D(\mathbf{x}^{\text{gen}})] - \mathbb{E}[D(\mathbf{x}^{\text{real}})] \tag{1}$$

For DOG, we reuse these loss functions (with batch averaging instead of expectation), but replace sample generation via a generator with a generation optimization (GO) process. GO aims to generate samples $\mathbf{x}^{\text{gen}}$ that (locally) minimize $L_G$, akin to samples from a GAN generator. As there is no generator model $G$ in DOG, we refer to $L_G$ as a "generation" loss rather than a "generator" loss. GO depends on the current discriminator $D$ (initialized randomly before training) and a random starting sample $\mathbf{x}_0^{\text{gen}} \sim \mathcal{P} = \mathcal{N}(\mathbf{0}, \mathbf{I})$. We iteratively generate a sample $\mathbf{x}^{\text{gen}} = \mathbf{x}_T^{\text{gen}} = \text{GO}(D, \mathbf{x}_0^{\text{gen}})$ using the gradient of $L_G$ with respect to $\mathbf{x}_t^{\text{gen}}$. The sequence of samples $\mathbf{x}_t^{\text{gen}}$ forms a GO path. We optimize for $T$ steps using gradient descent with a learning rate $\eta$:

$$\mathbf{x}_{t+1}^{\text{gen}} = \mathbf{x}_t^{\text{gen}} - \eta \nabla_{\mathbf{x}_t^{\text{gen}}} L_G(\mathbf{x}_t^{\text{gen}}) \tag{2}$$

Using only the final $\mathbf{x}^{\text{gen}}$ and a $\mathbf{x}^{\text{real}}$, we train $D$ to minimize the discriminator loss $L_D(\mathbf{x}^{\text{real}}, \text{sg}(\mathbf{x}^{\text{gen}}))$, where sg stops the gradient. As outlined in Algorithm 1, this process is repeated for each training step.

We stop the gradient to prevent collapse. Suppose we retained the gradients from GO for the discriminator weight update step and also required gradient computation for the weights of $D$. In that case, the computational graph through which we backpropagate for a training step would encompass all the $T$ forward and backward passes through $D$ from GO. This would lead to memory issues. Additionally, since $L_D$ compels $D$ to assign a lower score to the final generated sample $\mathbf{x}_T^{\text{gen}}$, $D$ would have an advantage in altering its score surface to eliminate GO paths from noise to realistic samples. For instance, by giving zero gradient to noisy inputs, the generated samples would remain noisy and $D$ could easily assign a low score to them. Therefore, we do not consider any intermediate samples $\mathbf{x}_t^{\text{gen}}$ ($t < T$) or their gradients for updating $D$. $D$ is also not conditioned on any timestep $t$, and the same $T$ is used for both training and inference.

DOG provides flexibility in the choice of $\mathcal{P}$, $L_G$, and $L_D$. In practice, we may also incorporate learning rate scheduling in GO, and Equation (2) can be extended for advanced optimizers.

## 3 CONVERGENCE ANALYSIS

To gain further insight, inspired by (Mescheder et al., 2018), we provide a convergence analysis of DOG using a straightforward dataset and a simple $D$. We show that in this setting, GO converges to local minima of $L_G$ (inner convergence), and that the discriminator training also converges (outer convergence). Additionally, since EBMs are conceptually similar to DOG, we elucidate how an EBM would falter here. For an experimental validation of the convergence analysis and a formal examination of DOG, with a discussion of our assumptions, see Appendix A.

We consider an underlying data distribution where samples lie on a regular 1-D grid, i.e., $x^{\text{real}} \in \{kn \mid n \in \mathbb{Z}\}$ (with a fixed scalar $k$). Suppose our training set only contains two real samples: $x_0^{\text{real}} = 0$ and $x_1^{\text{real}} = 2\pi$. For the discriminator, we select $D(x) = \cos(\theta x)$ with a single learnable scalar parameter $\theta$. Assuming suitable hyperparameters $T$ and $\eta$, GO converges to a local minimum of $L_G$ (maximum of $D$) from any starting sample $x_0^{\text{gen}}$, i.e., $x^{\text{gen}} \in \{2\pi n/\theta \mid n \in \mathbb{Z}\}$ such that $L_G(x^{\text{gen}}) = -1$ (with $D(x^{\text{gen}}) = 1$), showing inner convergence.

The expected value of the Wasserstein discriminator loss function can be computed as

$$\mathbb{E}[L_D] = \mathbb{E}[D(x^{\text{gen}})] - \mathbb{E}[D(x^{\text{real}})] \tag{3}$$
$$= \mathbb{E}[D(x^{\text{gen}})] - 0.5D(x_0^{\text{real}}) - 0.5D(x_1^{\text{real}}) \tag{4}$$
$$= \mathbb{E}[1] - 0.5D(0) - 0.5D(2\pi) \tag{5}$$
$$= 1 - 0.5 - 0.5\cos(\theta 2\pi). \tag{6}$$

Differentiating $\mathbb{E}[L_D]$ with respect to $\theta$ yields $d\mathbb{E}[L_D]/d\theta = \pi \sin(\theta 2\pi)$. Training with stochastic gradient descent on $\theta$ converges to $\theta = 1$ if an appropriate learning rate is used and $\theta$ is initialized sufficiently close, displaying outer convergence.

If we choose a broad distribution $\mathcal{P}$ for $x_0^{\text{gen}}$, DOG generalizes to the underlying grid data distribution because we can also generate samples that are not among the real samples. For instance, $x^{\text{gen}} = \text{GO}(D, 4\pi + \epsilon) = 4\pi$ for $x_0^{\text{gen}} = 4\pi + \epsilon$ and a small $\epsilon$. Note that other values, such as $\theta = 2$, would also minimize $L_D$ and can be reached by training with an appropriately initialized $\theta$. The choice of the discriminator and the initialization leads to different generalization patterns.

We could define an EBM by interpreting the model output as negative energy with corresponding density $p(x) \propto \exp(\cos(\theta x))$ (Du & Mordatch, 2019). However, such an EBM would fail for this setting: Even if the modes with high density lie on the grid-points, the density between the grid points is never zero. Therefore, any faithful sampling approach will also yield invalid samples between the grid-points. In contrast, DOG's GO only generates valid samples, showing an advantage of DOG over EBMs in this discrete setting.

## 4 EVALUATION

We evaluate DOG on datasets from multiple domains, including a 2D toy dataset with 25 Gaussians (Tran et al., 2018) and several graph datasets such as QM9, a molecule dataset (Ramakrishnan et al., 2014). For a proof of concept with limited generation quality in the image domain, see Appendix B.

Based on the results of early experiments, we generally use Adam (Kingma & Ba, 2015) as the optimizer for the GO process with $T = 100$ and a OneCycle learning rate schedule (Smith & Topin, 2019) (max_lr $= 1.0$ and a warm-up ratio of $0.3$) for faster convergence compared to constant learning rate gradient descent. An implementation can be found in the supplementary material.

### 4.1 25-GAUSSIANS TOY DATASET

To generate a 25-Gaussians dataset, we independently draw 2000 samples from a Gaussian with covariance $\mathbf{\Sigma} = \mathbf{I} \times 0.0001$ for each mean $\mu$ in $[-1.5, -0.75, 0, 0.75, 1.5]^2$ to obtain $50,000$ samples, similar to Tran et al. (2018). As in Wang et al. (2022), we use a discriminator with 2 hidden linear layers of 128 units each and LeakyReLU activations between them. We set max_lr $= 0.1$ for the OneCycle in the GO process and train with a batch size of 128 for 50 epochs using Adam with lr $= 10^{-5}$, $(\beta_1, \beta_2) = (0, 0.9)$, and Wasserstein losses (Arjovsky et al., 2017).

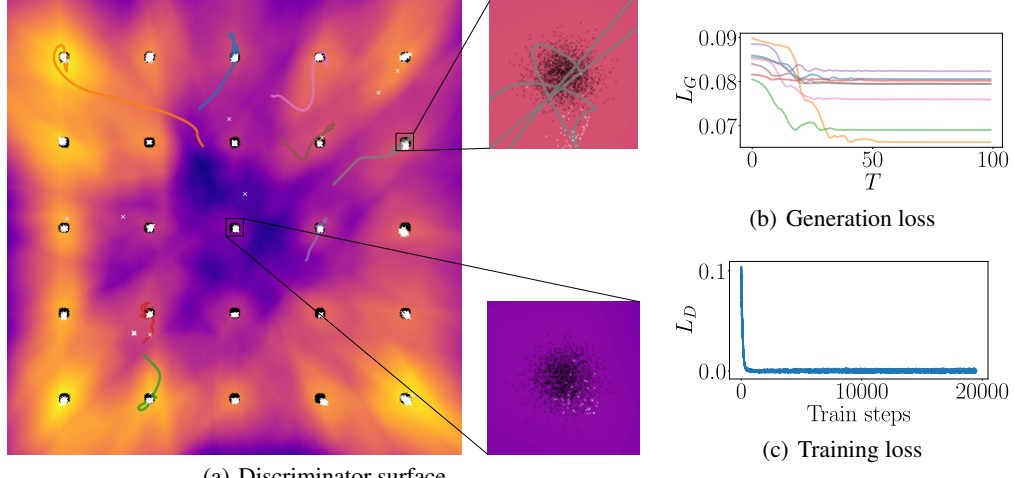

(a) Discriminator surface

(b) Generation loss

(c) Training loss

Figure 2: Results of DOG on the 25-Gaussians toy dataset. (a) The background colors represent the discriminator scores, with brighter colors indicating higher scores. The black crosses represent the $50,000$ real samples, while the white crosses represent $1,280$ generated samples. The colored lines show a subset of the GO paths, which start at random locations $\mathbf{x}_0^{\text{gen}}$ and end at the generated samples $\mathbf{x}_T^{\text{gen}}$. (b) Typical GO loss curves, colored according to the paths shown in (a). Both the initial and final scores vary, and all GOs have converged. (c) Discriminator training loss of 25-Gaussians for DOG. A loss close to zero indicates that generated and real samples receive similar scores.

As shown in Figure 2, DOG covers all modes of the data, and most of the generated samples are close to the real modes. Note that the discriminator scores (and the resulting loss surface) provide numerous GO paths to each mode, even though they are not perfectly symmetric, and some modes receive higher scores than others. In particular, the central mode at $(0, 0)$ is still covered, even though it receives a lower score than others, since GO often ends at local maxima. Furthermore, since the starting point density is $\mathcal{N}(\mathbf{0}, \mathbf{I})$, which is highest at $(0, 0)$, we already initialize many GO paths close to this mode.

## 4.2 GRAPH DATA

To generate graphs with DOG, we directly optimize both the node feature matrix and the edge feature matrix (or adjacency matrix in the absence of edge features). After each generation step in GO, we apply constraints to the edge matrix by setting its diagonals to zero and ensuring symmetry. We also mask the edge matrix and the node feature matrix based on the given number of nodes for each batch element. This is similar to the post-processing step used by Martinkus et al. (2022) for SPECTRE, which is the current state-of-the-art for graph generation using GANs.

**Model** Although DOG could use the more advanced discriminator of SPECTRE, which uses conditioning on eigenvalues and eigenvectors generated by a complex multi-step generator, for simplicity, we opt for the simple discriminator architecture of GG-GAN (Krawczuk et al., 2021). Specifically, we use the GG-GAN (RS)* reference implementation, introduced by Martinkus et al. (2022), with default hyperparameters. For example, for QM9, we use a discriminator with 3 PPGN layers (Maron et al., 2019), each with 64 channels, and a batch size of $128$.

**Data** For a detailed description of each dataset used for evaluation, data splits, and statistics, we refer to Martinkus et al. (2022). In short, we evaluate DOG on all the benchmark datasets on which SPECTRE was evaluated, including both artificial and real-world graphs. The artificial datasets are Community-small (You et al., 2018), with 12-20 nodes; SBM, which includes graphs sampled from stochastic block models; and Planar, containing only planar graphs. The two real-world datasets are Proteins (Dobson & Doig, 2003), containing proteins with hundreds of nodes, and QM9 (Ramakrishnan et al., 2014), an exhaustive dataset of molecule graphs with up to 9 heavy atoms.

**Metrics** Following Martinkus et al. (2022), we generate as many graphs for each dataset as there are in the test set. For the set of generated graphs, we report the percentage of valid, unique, and novel (V.,

Table 1: QM9 results. °As discussed by Vignac et al. (2023), novelty is a problematic metric for QM9. Therefore, we report it here only for completeness.

| METHOD | V.↑ | V. & U. ↑ | V., U.& N.° |
|---|---|---|---|
| DIGRESS | **99.0** | ≈95.2 | - |
| SPECTRE | 87.3 | 31.2 | 29.1 |
| GG-GAN (RS)* | 51.2 | 24.4 | 24.4 |
| DOG 3L. 64 CH. | 93.8±1.3 | 91.7±0.9 | 58.1±2.4 |
| DOG 6L. 128 CH. | 98.9 | **95.8** | 42.0 |

Figure 3: Uncurated set of DOG generated samples for QM9.

U. & N.) graphs, as well as the maximum mean discrepancy (MMD) for various statistics (Degree, Clustering, Orbit, Spectrum, and Wavelet) between the generated set and the test set, where applicable. We also report the average ratio among all statistics between the MMD of the generated set and the MMD of the training set, where a ratio of $1.0$ indicates that the generated set is as distinguishable from the test set as the training set. For Community-small, we use the earth mover's distance (EMD) instead of MMD to maintain consistency with Martinkus et al. (2022).

**Baselines** As baselines, we compare the performance of DOG with those of SPECTRE and GG-GAN (RS)*, as reported by Martinkus et al. (2022). Note that they use the node number distribution of the test set to generate samples, which we also adopt for consistency. Additionally, we include GG-GAN* or MolGAN* (De Cao & Kipf, 2018; Martinkus et al., 2022) where they perform better. Where available, we also include the results of DiGress (Vignac et al., 2023), a recent diffusion-based approach that does not use a discriminator and updates the graph in discrete steps. Note that Vignac et al. (2023) report the MMD ratios directly instead of the raw MMD values. Therefore, we calculate

Table 2: Community-small results. We explain the extraordinarily small ratio of EMD scores by the small number of test samples (and thus generated samples), as well as the fact that we use the node distribution of the test set to be consistent with Martinkus et al. (2022). Our results, as well as those of DiGress indicate that the performance is saturated for this toy dataset.

| METHOD | DEG. ↓ | CLUS. ↓ | ORBIT ↓ | RATIO ↓ |
|---|---|---|---|---|
| DIGRESS | ≈0.018 | ≈0.0643 | ≈0.006 | 1.0 |
| SPECTRE | 0.02 | 0.21 | 0.01 | 1.7 |
| GG-GAN (RS)* | 0.08 | 0.22 | 0.08 | 5.5 |
| DOG (OURS) | **0.003** | **0.006** | **0.002** | **0.16** |

them using the training MMD values of Martinkus et al. (2022), which may introduce rounding errors. The comparability of the results is further limited because DiGress uses different splits for some datasets and uses the node distribution of the training set for sampling.

**Other Settings** Like Martinkus et al. (2022), we train for $30$ epochs for QM9 and $12,000$ epochs for Community-small and Planar. Due to the expensive GO, we use only $130$ epochs for Proteins (instead of $1,020$) and $2,400$ for SBM (instead of $6,000$). Each run uses a single Nvidia A40, except for Proteins where we use 2 GPUs (and maintain a total batch size of 4 as in SPECTRE). Training on our default QM9 configuration takes about 15 hours. We keep the seeds constant as in the reference implementation and, like Martinkus et al. (2022), select a checkpoint for testing according to the validation performance. As the validation metric, we use the mean MMD ratio, except for QM9, where we choose the ratio of valid and unique generated graphs. Like Martinkus et al. (2022), we use WGAN-LP ($\lambda_{LP} = 5$) and gradient penalty as losses (Petzka et al., 2018). We train using Adam with lr $= 10^{-4}$ and $(\beta_1, \beta_2) = (0.5, 0.9)$.

**Results** As shown in Table 1, DOG outperforms all GAN approaches on QM9, including GG-GAN (RS)* and SPECTRE, while being competitive with DiGress when using a larger discriminator (6 layers, 128 channels each), closing the substantial performance gap between diffusion models and discriminator-based GANs. For our main experiment with the default configuration, we report the mean and standard deviation for five seeds. Examples of generated graphs are provided in Figure 3.

The results for other graph datasets also demonstrate DOG's superior graph generation performance over GANs, as can be observed in Table 2 (Community-small), Table 3 (Planar), Table 4 (SBM, max_lr $= 0.1$), and Table 5 (Proteins, max_lr $= 0.1$). Notably, DOG achieves this high performance

Table 3: Planar results.

| METHOD | DEG. ↓ | CLUS. ↓ | ORBIT ↓ | SPEC. ↓ | WAVELET ↓ | RATIO ↓ | V. ↑ | U. ↑ | N. ↑ | V., U. & N. ↑ |
|---|---|---|---|---|---|---|---|---|---|---|
| DIGRESS | ≈0.00028 | ≈**0.0372** | ≈0.00085 | - | - | - | - | - | - | **75** |
| SPECTRE | 0.0005 | 0.0785 | 0.0012 | 0.0112 | 0.0059 | 2.9 | 25.0 | 100.0 | 100.0 | 25.0 |
| GG-GAN (RS)* | 0.1005 | 0.2571 | 1.0313 | 0.2040 | 0.3829 | 586.3 | 0.0 | 100.0 | 100.0 | 0.0 |
| DOG (OURS) | **0.00023** | 0.0827 | 0.0034 | **0.0067** | **0.0029** | **2.78** | **67.5** | 100.0 | 100.0 | 67.5 |

Table 4: SBM results.

| METHOD | DEG. ↓ | CLUS. ↓ | ORBIT ↓ | SPEC. ↓ | WAVELET ↓ | RATIO ↓ | V. ↑ | U. ↑ | N. ↑ | V., U. & N. ↑ |
|---|---|---|---|---|---|---|---|---|---|---|
| DIGRESS | ≈0.00128 | ≈**0.0498** | ≈0.04335 | - | - | - | - | - | - | **74** |
| SPECTRE | 0.0015 | 0.0521 | 0.0412 | 0.0056 | 0.0028 | 2.0 | 52.5 | 100.0 | 100.0 | 52.5 |
| GG-GAN (RS)* | 0.0338 | 0.0581 | 0.1019 | 0.0613 | 0.1749 | 61.5 | 0.0 | 100.0 | 100.0 | 0.0 |
| GG-GAN* | 0.0035 | 0.0699 | 0.0587 | 0.0094 | 0.0202 | 7.8 | 25.0 | 100.0 | 100.0 | 25.0 |
| DOG (OURS) | **0.0003** | 0.0508 | **0.0401** | **0.0039** | **0.0013** | **1.15** | **72.5** | 100.0 | 100.0 | 72.5 |

Table 5: Proteins results. °Note that novelty is severely limited for MolGAN* as discussed by Martinkus et al. (2022).

| METHOD | DEG. ↓ | CLUS. ↓ | ORBIT ↓ | SPEC. ↓ | WAVELET ↓ | RATIO ↓ | U. ↑ | N. ↑ | U. & N. ↑ |
|---|---|---|---|---|---|---|---|---|---|
| SPECTRE | 0.0056 | 0.0843 | 0.0267 | 0.0052 | 0.0118 | 16.9 | **100.0** | **100.0** | **100.0** |
| GG-GAN (RS)* | 0.4727 | 0.1772 | 0.7326 | 0.4102 | 0.6278 | 875.8 | **100.0** | **100.0** | **100.0** |
| MOLGAN* | **0.0008** | **0.0644** | **0.0081** | **0.0021** | **0.0012** | **4.2** | 97.3 | 100.0° | 97.3 |
| DOG (OURS) | 0.0022 | 0.0682 | 0.0202 | 0.0014 | 0.0023 | 6.75 | **100.0** | **100.0** | **100.0** |

using only the simple GG-GAN (RS)* discriminator, and we did not tune any GAN-specific hyperparameters. We only tuned the DOG-specific max_lr and kept $T = 100$ constant. For examples of both real and generated graphs, see Figure 10 (Appendix).

**Hyperparameter Study** We evaluate the impact of our design choices on DOG's performance on QM9. In Table 6, we compare against the default model w.r.t. the fraction of valid and unique graphs. We also analyze the discriminator's smoothed Wasserstein loss $L_D$ at the end of training, as we observed in early experiments that a low loss correlates with poor samples. This suggests that it is undesirable for GO if the discriminator can easily distinguish between generated and real samples.

First, we investigate the model size by doubling the number of channels and layers, both independently and together. Models larger than those used by the GAN baselines perform better.

Next, we examine our choices for GO: While increasing the number of steps $T$ may result in better convergence to local minima of $L_G$ (maxima in $D$'s score surface), it also increases the computational cost of DOG, posing a trade-off. Doubling the number of steps to $T = 200$

Table 6: Hyperparameter study for DOG on QM9. Each change to the default setting is indicated in the first column. Default: $T = 100$, OneCycle (max_lr = 1.0), Adam, 3 layers, 64 channels, WGAN-LP losses. °Not comparable because different loss functions or number of epochs are used.

| SETTING | V. & U. ↑ | FINAL $L_D$ |
|---|---|---|
| DEFAULT DOG | $91.7_{\pm 0.9}$ | $-0.17_{\pm 0.04}$ |
| 128 CH. | 93.8 | -0.08 |
| 6 LAY. | 94.3 | -0.21 |
| 128 CH., 6 LAY. | **95.8** | -0.14 |
| T=200 | 92.0 | -0.07 |
| T=50 | 81.3 | -0.79 |
| T=25 | 34.1 | -44 |
| MAX_LR = 0.1 | 90.2 | -0.29 |
| ADAM → SGD | 91.1 | -0.16 |
| NO ONECYCLE | 56.9 | -15 |
| NO CONSTRAIN | 81.8 | -0.34 |
| NS LOSSES | 84.4 | 1.27° |
| 1 EPOCH TRAINING | 43.4 | -0.04° |

results in higher discriminator losses but similar quality, indicating that $T = 100$ is sufficient. In contrast, halving the number of steps to $T = 50$ leads to worse performance. An even lower number of steps ($T = 25$) leads to a failure mode where GO ends before the discriminator can be fooled, giving the discriminator a low loss. Reducing the learning rate in GO by a factor of 10 leads to only slightly worse results, indicating that GO is not overly sensitive to learning rates, at least when using Adam with OneCycle learning rate scheduling as the optimizer. However, not using learning rate scheduling (but tuned lr = 0.1) significantly reduces the sample quality, suggesting that learning rate scheduling is crucial for GO, as again implied by the low discriminator loss. Replacing Adam with stochastic gradient descent (SGD) in the GO process performs similarly to the default setting but requires more learning rate tuning (max_lr = 100). Not constraining the edge matrix after each GO step has a negative effect, demonstrating the benefit of staying within the data domain during GO.

Replacing WGAN losses with non-saturating (NS) GAN losses (Goodfellow et al., 2014; Karras et al., 2019) (but keeping $\lambda_{LP}$ and other hyperparameters) leads to only a moderate degradation, supporting our claim that DOG, like GANs (Qin et al., 2018), is flexible w.r.t. the choice of $L_D$ and $L_G$.

While individual training steps of DOG take more time due to the iterative GO, we find that the overall training nevertheless progresses faster compared to SPECTRE. Although SPECTRE trains by default for 30 epochs ($\approx 8$ hours) on QM9, DOG achieves better test performance after only one epoch ($< 0.5$ hours), indicating faster convergence.

## 5 RELATED WORK

**GANs & EBMs** From a GAN perspective (Goodfellow et al., 2014), DOG can be obtained by removing the generator model and, to generate samples, replacing it with GO, a multi-step gradient-based optimization utilizing the current discriminator. For training, the generator optimization step is removed without replacement, as only one model, the discriminator, remains. In the discriminator update step, the generated samples and real samples are processed in the same way as for GANs.

In EBMs (Ackley et al., 1985; Xie et al., 2016; 2017; Du & Mordatch, 2019), an energy estimation model is trained to assign low energy scores to real samples and high energy scores to generated samples, essentially using a Wasserstein loss (Arjovsky et al., 2017). A main difference between DOG and EBMs lies in the sampling procedure, particularly in the absence of noise. EBMs interpret the model's output as energy that determines the (unnormalized) density at each location. An ideal sampler would therefore draw samples from this density. An example of such a sampler is Langevin dynamics (LD) (Welling & Teh, 2011), which utilize noise and the gradient of the energy score with respect to the input sample to update the input sample in several steps. Potentially, such a sampling approach can be paired with a sample replay buffer (Du & Mordatch, 2019) to facilitate faster mixing of the underlying Markov chain and therefore save expensive generation steps during training. In practice, however, the sampling from the density is often not perfect: LD may use a limited number of steps (Nijkamp et al., 2019), or smaller noise than theory requires (Xiao et al., 2021), introducing a distribution shift. However, without any noise in the LD (Xiao et al. (2021) term this "EBMs w/ noise-free dynamics" and Nijkamp et al. (2019) argue that the gradient dominates the noise in certain settings), we perform gradient descent on the energy model's input and, assuming convergence, only obtain samples that lie at local minima of the energy surface. Therefore, we no longer attempt to acquire samples accurately following the energy-defined density. This minimization is akin to minimizing a generation loss with Wasserstein GAN losses, as also pointed out by Xiao et al. (2021). Due to this difference in acquired samples, we abandon the "EBMs w/ noise-free dynamics" (Xiao et al., 2021) interpretation and instead consequently interpret it as a special case of DOG.

This shift in interpretation from imperfect sampling from an energy-defined density via impaired LD (or any other technique) to optimizing the sample by locally minimizing a generation loss (maximizing a discriminator's score) motivates the other distinctions between EBMs and DOG: Namely, we can employ tools developed for efficient gradient-based optimization, such as momentum-based optimizers and learning rate schedulers. Also, GAN losses, discriminator architectures, regularization techniques such as gradient penalty and other hyperparameters largely transfer, as demonstrated in our graph experiments. Furthermore, local minima can possess the same sampling density even if they receive different absolute scores from the discriminator/energy model (refer to Figure 2). Finally, a sample replay buffer is not required since DOG aims to converge to a local minimum instead of striving for a mixed Markov chain.

Overall, DOG combines concepts from both GANs and EBMs. It repurposes GAN loss functions and discriminator architectures but generates samples iteratively using a single scalar prediction model like EBMs. Unlike GANs, DOG does not require a separate generator architecture and training process, and unlike EBMs, it does not involve sample replay buffers for training or random noise in the generation process. Instead, DOG employs direct gradient descent with respect to a generation loss, potentially combined with momentum-based optimizers and learning rate scheduling.

EBMs and GANs are also related. While they can be combined (Zhao et al., 2017; Pang et al., 2020), Che et al. (2020) show how a GAN can be interpreted as an EBM, and Xiao et al. (2021) indicate that EBMs are self-adversarial like DOG. Additionally, Tanaka (2019) and Ansari et al.

(2020) demonstrate how to use the gradient of a discriminator to refine the generation, which may come from a GAN generator.

**Other Generative Modeling Approaches** Other approaches related to EBMs and DOG include score-based generative modeling (Song & Ermon, 2019; Song et al., 2021) and denoising diffusion approaches (Sohl-Dickstein et al., 2015; Ho et al., 2020). All of these methods generate samples incrementally by making dense predictions on the input, starting from noise. They allow for corrections of previous inaccuracies during generation, unlike GANs, which generally generate samples in a one-shot fashion. Diffusion models typically predict the remaining noise, while score-based models estimate the gradient of the energy surface. Unlike EBMs and DOG, the dense prediction in these methods does not come from a backward pass through a scalar prediction model but from a forward pass through a dense prediction model. In addition, these settings do not require expensive iterative sample generation during training but cheaply noise real samples. For inference, the updates of denoising diffusion follow a pre-determined schedule similar to a learning rate scheduling in GO.

Further, Diffusion-GAN (Wang et al., 2022) and discriminator guidance (Kim et al., 2022) combine ideas from diffusion and GANs by using a discriminator to refine partially denoised samples generated by a diffusion model. Complementary to our work, Franceschi et al. (2023) provide a formal connection between GANs and score-based diffusion. DOG is similar to a particle model. However, the DOG discriminator is not conditioned on timesteps, and the samples may be updated with momentum and learning rate scheduling instead of just the gradients. In contrast to DOG, their Discriminator Flow approach requires this conditioning for the discriminator and uses the intermediate samples for training, making DOG conceptually simpler.

Besides these, a range of other popular generative approaches have been explored, including normalizing flows (Rezende & Mohamed, 2015; Kobyzev et al., 2019; Xie et al., 2022), variational autoencoders (Kingma & Welling, 2014; Vahdat & Kautz, 2020), autoregressive generation (Brown et al., 2020; Lee et al., 2022), and masked modeling (Chang et al., 2022; 2023).

**Adversarial Attacks** DOG is also related to adversarial robustness (Song et al., 2018; Dong et al., 2020): Both settings use the gradient of a model's input to perturb samples, resulting in a change in the model's predictions. GO is essentially an unrestricted adversarial attack on the discriminator $D$ starting at noise, as it aims to maximize $D$'s scores. $D$ is trained to be robust to these attacks. However, the goal of DOG is different: it aims at de novo content generation and receives only noise as input during inference, no adversarially perturbed samples. In contrast, adversarial reconstruction attacks (Balle et al., 2021; Haim et al., 2022) aim to generate realistic samples, like DOG. While reconstruction attacks aim to retrieve training samples, DOG's goal is to generalize.

**Graph Generation** Specifically for the graph domain, besides the multi-step GAN-based SPECTRE (Martinkus et al., 2022) and the state-of-the-art diffusion-based approaches such as DiGress (Vignac et al., 2023) or follow-ups (Chen et al., 2023; Kong et al., 2023), other methods have been proposed. These include score-based generative modeling (Niu et al., 2020; Jo et al., 2022) and autoregressive models (Liao et al., 2019). Earlier graph GANs are based on recursive neural networks (You et al., 2018) or random walks (Bojchevski et al., 2018). Other approaches utilize normalizing flows (Madhawa et al., 2020) or focus on permutation invariance (Vignac & Frossard, 2022). While these approaches are tailored to graphs, sometimes even limited to molecule graphs, DOG is a more general approach that works well on graphs.

## 6 DISCUSSION

**Training Efficiency** DOG outperforms approaches like SPECTRE (Martinkus et al., 2022) on graphs without requiring an elaborate multi-step approach where some generators and discriminators need to be trained before others. This greatly simplifies the setting by using only a single discriminator model. Moreover, unlike the extensive journey of different generators from GraphGAN (Wang et al., 2018) to SPECTRE, there is no need to tune the generator architecture for DOG.

Furthermore, DOG has an advantage over DiGress in that it requires fewer parameters and significantly fewer training epochs (Vignac et al., 2023). For QM9, DiGress was trained for 1000 epochs using an expensive transformer model with 4.6 million parameters. In contrast, DOG achieves the same sample quality with only 30 epochs and 1.1 million parameters for the large discriminator configuration (and merely 145 thousand parameters for the default configuration).

Another conceptual advantage of DOG is its ability to instantly update GO with $D$, directly taking advantage of the adjusted discriminator weights from the previous training step, potentially providing high-quality training samples (namely those that maximally fool $D$) to $D$ early on. This is in contrast to GANs, where the generator model must first learn from the discriminator and always lags behind. As a result, DOG may require fewer training steps to achieve good performance, as demonstrated on QM9 (see Section 4.2). This observation suggests that in DOG, inner (generation) and outer (training) optimization are closely related.

Nevertheless, while DOG shows advantages over GANs on graphs regarding the outer optimization, one of its main limitations is the currently expensive training steps that include GO in the inner optimization. A detailed analysis of the computational cost can be found in Appendix C.

A potential solution to improve training efficiency is to reuse generated samples during training. By using a sample replay buffer, such as (Du & Mordatch, 2019), the starting point for GO could be closer to realistic samples, thus requiring fewer GO steps to achieve convergence. Another approach could be to use the intermediate samples and their gradients, instead of discarding them altogether with the stop gradient operator, to regularize and potentially shorten the GO paths. Additionally, we expect that more hyperparameter tuning for Adam and OneCycle, or using a more suitable GO approach altogether, could further reduce the number of steps required. By analogy, note that earlier denoising diffusion approaches used thousands of steps, which could be reduced to only a handful by skipping steps and increasing the step size (Meng et al., 2022).

**GO Paths** While DOG is seemingly more parameter-efficient than GANs since it does not require the parameters of a generator, the DOG discriminator has a different task to solve. Given a broad distribution $\mathcal{P}$, GO can start anywhere and, assuming appropriate hyperparameters, will eventually reach all local maxima in the score surface of $D$. Thus, a good discriminator should have local maxima only at realistic samples and should be robust to the adversarial attack GO performs. This observation suggests that the score surface of a DOG discriminator must provide meaningful gradients over the entire target domain for the GO path from each noisy sample to a realistic one. This is in contrast to the score surface of a GAN discriminator, which only needs to provide meaningful gradients on the subset of the domain currently covered by the generator. The generator is restricted by its architecture and its current parameters and therefore many local maxima in the GAN discriminator's score surface are not sampled and $L_D$ is not computed at these locations. For example, for face image data, after teaching a generator the global structure of a face, a GAN discriminator could adapt to focus on the local texture but struggle with less structured data (as can be seen in Appendix B.1). Therefore, we speculate that a DOG discriminator might benefit from an architecture and regularization techniques that are optimized to accommodate this difference.

Assuming good performance, where the distribution of generated samples largely follows the real data distribution, the GO paths must be meaningful. The emergence of these paths is non-trivial: it is not enough to have realistic samples at local maxima; they must also be reachable via the GO paths with an appropriate probability. In particular, the loss function $L_D$ only applies directly to the real and generated samples, not to distant locations (i.e., $\mathbf{x}_0^{\text{gen}}$). However, the gradient at these locations may already determine the general direction of many GO paths. For larger settings than the one described in Section 3, investigating how these paths emerge, such that the generated samples follow the underlying data distribution using only $L_D$, is left for future work. Potentially, these paths emerge in a manner similar to the paths taken by adversarial reconstruction attacks in classification models (Haim et al., 2022).

## 7 CONCLUSION

Overall, DOG's ability to provide high-quality training samples directly to the discriminator $D$, along with its simplified setting that eliminates the need to tune a generator architecture, makes it a promising generative approach. On graphs, we demonstrate that high-quality sample generation is possible with the conceptually simple self-adversarial approach, outperforming GANs while being on par with state-of-the-art diffusion approaches. Further improvements to the discriminator architecture, regularization techniques, and DOG's speed may also enable competitive performance in other domains.

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

# A    METHOD ANALYSIS

This section provides theoretical insights into DOG's workings. While we utilize Wasserstein GAN loss terms (Arjovsky et al., 2017) for simplicity, similar derivations are also applicable to other terms.

For Wasserstein losses, DOG's update rule (Equation (2)) can be simplified to:

$$\mathbf{x}_{t+1}^{\text{gen}} = \mathbf{x}_t^{\text{gen}} + \eta \nabla_{\mathbf{x}_t^{\text{gen}}} D(\mathbf{x}_t^{\text{gen}}) \tag{7}$$

## A.1    MIN-MAX OPTIMIZATION

To better understand DOG from the perspective of saddle-point min-max optimization (Yadav et al., 2018), we consider the Wasserstein GAN. The optimization problem is defined with $\psi$ and $\boldsymbol{\theta}$ representing the generator and discriminator parameters respectively, and $\mathcal{Z}$ and $\mathcal{R}$ representing the latent and real data distribution respectively:

$$\min_{\psi} \max_{\boldsymbol{\theta}} \mathbb{E}_{\mathbf{x} \sim \mathcal{R}}[D_{\boldsymbol{\theta}}(\mathbf{x})] - \mathbb{E}_{\mathbf{z} \sim \mathcal{Z}}[D_{\boldsymbol{\theta}}(G_{\psi}(\mathbf{z}))] \tag{8}$$

We could replace $G$ with a perfect adversary of $D$, which always finds a global maximum in the score surface of $D$ (global minimum for $L_G$). This adversary thus acts as a better $G$, directly knowing everything that $D$ has learned, without being limited by its inductive biases such as model architecture:

$$\max_{\boldsymbol{\theta}} \mathbb{E}_{\mathbf{x} \sim \mathcal{R}}[D_{\boldsymbol{\theta}}(\mathbf{x})] - D_{\boldsymbol{\theta}}(\text{argmax}_{\mathbf{x}} D_{\boldsymbol{\theta}}(\mathbf{x})) \tag{9}$$

However, perfect adversaries are not practical, hence we use GO to approximate $\text{argmax}_{\mathbf{x}} D_{\boldsymbol{\theta}}(\mathbf{x})$:

$$\max_{\boldsymbol{\theta}} \mathbb{E}_{\mathbf{x} \sim \mathcal{R}}[D_{\boldsymbol{\theta}}(\mathbf{x})] - \mathbb{E}_{\mathbf{x}_0^{\text{gen}} \sim \mathcal{P}}[D_{\boldsymbol{\theta}}(\text{sg}(\text{GO}(D_{\boldsymbol{\theta}}, \mathbf{x}_0^{\text{gen}})))] \tag{10}$$

In practice, we alternate between obtaining a $\mathbf{x}^{\text{gen}} = \text{GO}(D_{\boldsymbol{\theta}}, \mathbf{x}_0^{\text{gen}})$ and optimizing $\boldsymbol{\theta}$ with a gradient descent step. As described in Section 2, during training, we must prevent gradient computation through GO to ensure that GO remains a good approximator for the argmax.

## A.2    SCORE SURFACE

Consider the score surface defined by $D$ (see Figure 2(a)) to gain another perspective. Assuming a smooth surface and converging GO, the generated samples lie at local maxima of the score surface (local minima of the $L_G$ surface). The gradient from $L_D$ on $D$'s parameters increases the scores of the real samples, potentially creating new local maxima, while it decreases the scores of the generated samples, potentially destroying their corresponding local maxima. While this process could introduce new local maxima elsewhere, which might then be destroyed again, consider a situation where there is exactly one strict local maximum at each real sample, and no other local maxima after training. This means that $D$ has memorized every sample in the training set since, by assumption, GO always ends at local maxima.

Note that GAN optimization seeks a Nash equilibrium (Heusel et al., 2017) between $G$ and $D$, where neither has an advantage by changing its weights. Suppose further that all local maxima are sampled with equal probability for DOG, i.e., the real and generated samples are drawn from the same uniform distribution $\mathcal{R}$ over the set of real training samples. In this situation, we would have reached a Nash equilibrium between GO and $D$: In this game, only one player, $D$, can choose its action (by changing its weights), and the other player, GO, is fixed given $D$ as it has no learnable parameters and therefore cannot gain an advantage. For $D$, the terms of $L_D$ cancel out and there is no incentive to change its weights. For example, for Wasserstein losses, we have, in expectation,

$$L_D = \mathbb{E}_{\mathbf{x}_0^{\text{gen}} \sim P}[D_{\boldsymbol{\theta}}(\text{sg}(\text{GO}(D_{\boldsymbol{\theta}}, \mathbf{x}_0^{\text{gen})}]))] - \mathbb{E}_{\mathbf{x}^{\text{real}} \sim \mathcal{R}}[D_{\boldsymbol{\theta}}(\mathbf{x}^{\text{real}})] \tag{11}$$

$$= \mathbb{E}_{\mathbf{x}^{\text{gen}} \sim \mathcal{R}}[D_{\boldsymbol{\theta}}(\mathbf{x}^{\text{gen}})] - \mathbb{E}_{\mathbf{x}^{\text{real}} \sim \mathcal{R}}[D_{\boldsymbol{\theta}}(\mathbf{x}^{\text{real}})] = 0 \tag{12}$$

and hence $\nabla_{\boldsymbol{\theta}} L_D = \mathbf{0}$.

However, in practice, pure memorization of the training samples is not desirable, as we want the distribution of the generated samples from $\text{GO}(D, \mathcal{P})$ to match the underlying distribution of the real

samples. We are interested in generating new samples from the domain and thus in generalization. For GANs, generalization comes from an imperfect discriminator or an imperfect generator (Zhang et al., 2018; Brock et al., 2019). Similarly, generalization in DOG must originate from an imperfect $D$ that also has other local maxima at other realistic samples from the domain (such as the one in Section 3) or from an imperfect GO that does not converge to local maxima but still ends at a realistic sample.

## A.3 DENSITY

We can obtain the density $p(\mathbf{x}^{\text{gen}})$ of a generated sample $\mathbf{x}^{\text{gen}}$ by counting the GO paths that end there and weighting them by the density of their start points $\mathbf{x}_0^{\text{gen}}$:

$$p(\mathbf{x}^{\text{gen}}) \propto \mathbb{E}_{\mathbf{x}_0^{\text{gen}} \sim \mathcal{N}(\mathbf{0},\mathbf{I})}[I(\text{GO}(D_{\boldsymbol{\theta}}, \mathbf{x}_0^{\text{gen}}), \mathbf{x}^{\text{gen}})] \tag{13}$$

Here, $I(a, b)$ is the indicator function that returns 1 if $a = b$ and 0 otherwise.

We define the set of local maxima of the discriminator $D$ as $\text{locmax}(D) := \{\mathbf{x} | \mathbf{x} \text{ is a local maximum of } D(\mathbf{x})\}$. Assuming that the gradient descent GO always converges to a local maximum $\mathbf{x} \in \text{locmax}(D)$, we can partition the space of starting locations according to the local maxima that are reached, turning the density $p$ into a probability if the set is discrete (i.e., if all local maxima are strict). Note, however, that $p(\mathbf{x})$ may be higher for some $\mathbf{x}$ than for others, meaning that more gradient descent paths (weighted by their starting density $\mathcal{P}(\mathbf{x}_0^{\text{gen}})$) end at some $\mathbf{x}$ than at others.

If we further assume that $D$ is smooth, all local maxima are strict, and $\eta$ is sufficiently small, then there exists a small radius $r$ around each $\mathbf{x} \in \text{locmax}(D)$ such that for every $\mathbf{x}_0^{\text{gen}}$ in this radius ($||\mathbf{x}_0^{\text{gen}} - x||_2 < r$), we have $I(\text{GO}(D, \mathbf{x}_0^{\text{gen}}), \mathbf{x}) = 1$, i.e., the GO paths lead to the closest local maximum. This implies that every local maximum will eventually be sampled, i.e., $\forall \mathbf{x} \in \text{locmax}(D) : p(\mathbf{x}) > 0$, since the starting distribution $\mathcal{P} = \mathcal{N}(\mathbf{0},\mathbf{I})$ is positive everywhere. This affects both training and inference: Only those local maxima that actually receive samples $\mathbf{x}^{\text{gen}}$ can be used by the loss function $L_D$ during training and are relevant for inference.

## A.4 ASSUMPTIONS

We will now assess the validity of the previously made assumptions.

Firstly, the assumption of smoothness holds for all discriminators that utilize differentiable (activation) functions. Furthermore, for discriminators that are almost everywhere differentiable, such as those using ReLUs, it is unlikely to encounter non-differentiable points due to numerical reasons.

Regarding the assumption of GO convergence, as long as $\eta$ is appropriately small and $T$ is sufficiently large, the GO process will come arbitrarily close to a local maximum in the score surface. In practice, however, we often empirically select $\eta$ and $T$ for good generation performance without necessarily ensuring full convergence.

Finally, we assume the absence of non-strict local maxima, which can be avoided by training discriminators with a gradient penalty (Gulrajani et al., 2017). With this technique, the gradients provided by the discriminator on its inputs maintain specific values almost everywhere, thus preventing the existence of flat regions required for non-strict local maxima.

While these assumptions are useful for theoretical analyses, they may not always hold true in practice. As discussed in Appendix A.2, an imperfect GO for example might even be a requirement for generalization.

## A.5 EMPIRICAL VALIDATION OF CONVERGENCE ANALYSIS

In practice, DOG's behavior on the 1D toy dataset aligns with the theory described in Section 3: We use $\mathcal{P} = \mathcal{N}(0, 1)$, $T = 100$, $\eta = 1.0$, and learning_rate_d $= 0.05$, a batch size of 2, and SGD for both the discriminator and GO. As expected, the generation loss always converges to $L_G = -1.0$ for GO, i.e., the generated samples lie at the local maxima of the cosine discriminator. The discriminator's parameter (initialized at $1.1$) converges to (numerically) $1.0$ in fewer

than 20 training steps: [1.100000023841858, 1.0261367559432983, 1.0158647298812866, 1.009611964225769, 1.009611964225769, 1.00581955909729, 1.0012255907058716, 1.000257968902588, 1.0001561641693115, 1.0001561641693115, 1.0000945329666138, 1.000019907951355, 1.0000041723251343, 1.0000008344650269, 1.0000004768371582, 1.000000238418579, 1.0000001192092896, 1.0, 1.0, 1.0, 1.0].

### A.6 Training Dynamics

Note that for successful DOG training runs, we are usually able to obtain samples with realness scores close to random guessing. For example, in Figure 2(c) and Table 6, the Wasserstein $L_D$ is close to zero but not substantially below it, indicating that the generated and real samples receive similar scores. If the generated samples were given low scores by the discriminator, the training would collapse because the discriminator would only be trained to assign low scores to samples it had already assigned low scores to before. However, if GO can find holes in the loss surface by achieving higher realness scores than real samples, there is an incentive for the discriminator to fill the holes by ignoring the input or by outputting a score close to random guessing.

These considerations further motivate the use of gradient penalty (Gulrajani et al., 2017) or spectral norms (Miyato et al., 2018) in the discriminator to avoid having zero gradients on its input for GO. In practice, we also observe that the loss of the DOG discriminator varies less compared to GAN training, which is potentially relevant for tuning the weights of regularization terms such as balanced consistency regularization (bCR) (Zhao et al., 2021).

## B DOG on Image Data

### B.1 Fixed GAN-trained Discriminator

Similar to early visualization techniques (Springenberg et al., 2015; Mordvintsev et al., 2015), we establish a baseline for DOG's performance by using a discriminator that was trained in the default StyleNAT (GAN) setting (Walton et al., 2022) without any further training for sampling with DOG's GO without constraints. The results can be observed in Figure 4 (Adam+OneCycle, $max\_lr = 1.0$, $T = 100$, other $max\_lr$ values lead to similar results). We see that a GAN-trained discriminator does not produce meaningful images, indicating that DOG's gradient-based GO struggles to find a path to a realistic sample when using a GAN-trained discriminator. However, this result is intriguing because it suggests that the DOG training process leads to a fundamentally different discriminator that enables GO paths to more realistic samples, as discussed in Section 6.

Following GAN training, the generator ideally generates high-quality samples. GAN discriminators are trained to differentiate real samples from those high-quality samples. They are not necessarily able to detect low-quality samples anymore because they did not see them as training samples recently (unlike earlier in training). Consequently, these discriminators might assign higher scores to such low-quality samples than to real or high-quality ones.

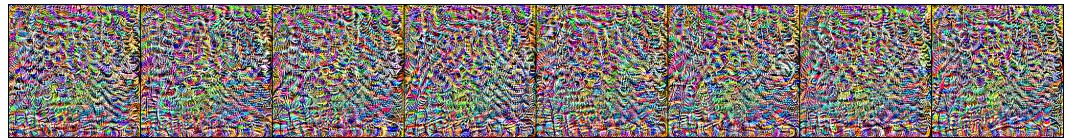

Figure 4: Generated samples for FFHQ using a (fixed) GAN-trained discriminator.

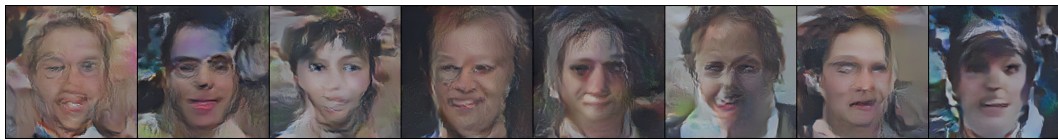

Figure 5: Generated samples for FFHQ using a DOG-trained discriminator.

## B.2 MODERN GAN DISCRIMINATOR

For a first demonstration of DOG on image data, we utilize FFHQ (256² pixels) (Karras et al., 2019) and largely follow the settings of StyleNAT (Walton et al., 2022). However, due to the high computational cost of DOG, we use a slimmer discriminator architecture with a quarter of the channels and train the model for only 100 epochs, which is about an order of magnitude less than StyleNAT. Other hyperparameters and regularization terms, such as NS GAN losses, horizontal flips for data augmentation, bCR, R1 regularization (Mescheder et al., 2018), a batch size of 64, and Adam with $\text{lr} = 2 * 10^{-4}$ and $(\beta_1, \beta_2) = (0, 0.9)$, remain the same. While an exponential moving average was used for the generator in the original setting, we do not apply it to the discriminator for DOG.

Although the displayed images in Figure 5 show faces, their quality is not competitive with the current state of the art, as quantified by a FID-50k of 142.86 for DOG versus 2.05 for StyleNAT. Since discriminators have only been used in conjunction with generators so far, they may implicitly exploit the generator's architecture.

## B.3 EBM BASELINE

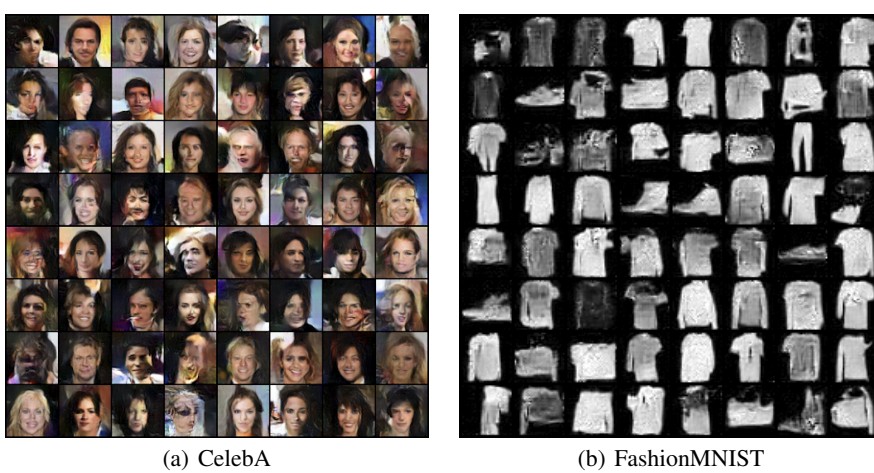

(a) CelebA  (b) FashionMNIST

Figure 6: Generated samples using DOG with EBM settings. For a qualitative comparison with EBMs, refer to (Xiao et al., 2021).

We report further experiments on smaller image datasets to compare DOG with EBMs. We use the settings from (Xiao et al., 2021) to compare DOG and EBMs on FashionMNIST (resized to 32x32) (Xiao et al., 2017) and CelebA (64x64) (Liu et al., 2015).
Specifically, we employ the

Table 7: FIDs ↓. EBM values from (Xiao et al., 2021).

|  | CELEBA | FASHIONMNIST |
|---|---|---|
| EBMS W/ NOISY DYNAMICS | 61.7 | 69.6 |
| EBMS W/ NOISE-FREE DYNAMICS | 50.1 | 56.6 |
| DOG | 53.2 | 53.3 |

same model architecture of the EBM as a discriminator, a batch size of 64, and Adam (Kingma & Ba, 2015) with a learning rate of $5 * 10^{-4}$. We train for $8,000$ and $30,000$ steps for FashionMNIST and CelebA, respectively. We utilize the same number of generation steps ($T = 40$ for FashionMNIST and $T = 60$ for CelebA) as their EBMs for the generation optimization (GO). We use Wasserstein losses as they yield gradients for the model parameters that are equivalent to the gradients in EBM training (Equation 1 in (Xiao et al., 2021)). For these experiments, we utilize SGD in GO and cosine annealing (Loshchilov & Hutter, 2016) as the learning rate scheduler. We tuned the learning rates for GO: $20.0$ for FashionMNIST and $0.5$ for CelebA. For the generated images, please refer to Figure 6. In Table 7, we present results from (Xiao et al., 2021) for EBMs ("EBM w/ noisy dynamics"), and

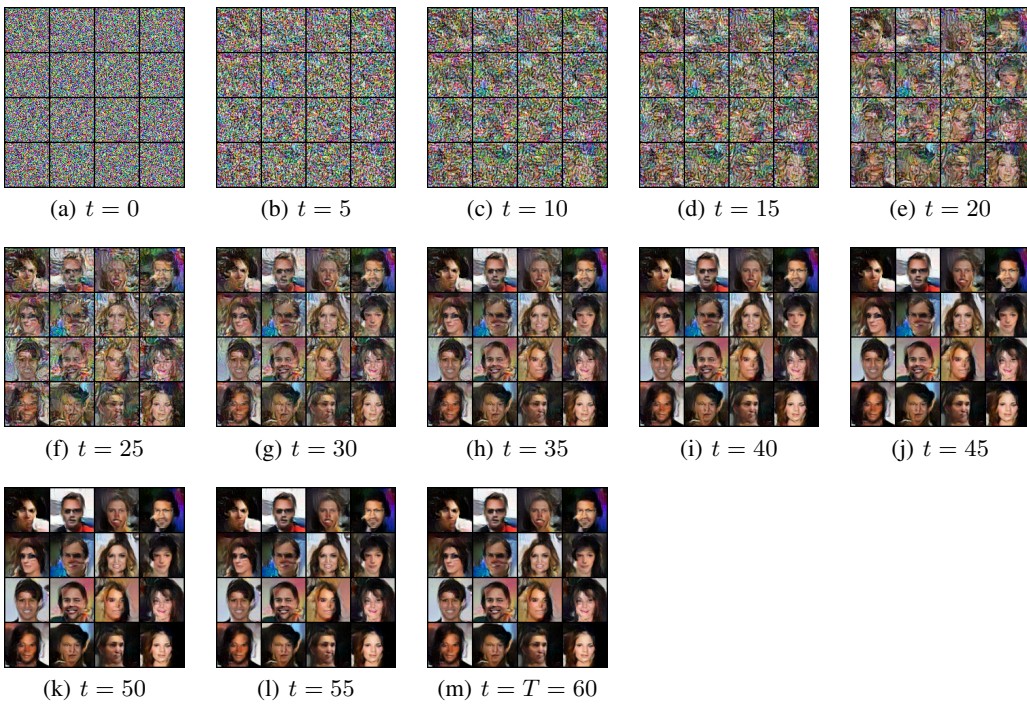

(a) $t = 0$   (b) $t = 5$   (c) $t = 10$   (d) $t = 15$   (e) $t = 20$

(f) $t = 25$   (g) $t = 30$   (h) $t = 35$   (i) $t = 40$   (j) $t = 45$

(k) $t = 50$   (l) $t = 55$   (m) $t = T = 60$

Figure 7: CelebA samples $\mathbf{x}_t^{\text{gen}}$ for different $t$.

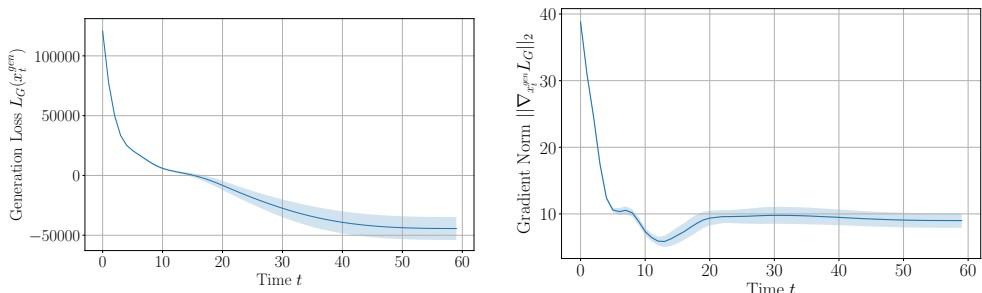

Figure 8: Generation loss and gradient norm for the CelebA samples $x_t^{\text{gen}}$ from Figure 7. Mean and standard deviation are calculated over the batch.

for comprehensiveness, we include results for "EBM w/ noise-free dynamics," which, as discussed in Section 5, can be seen as a special case of DOG (without learning rate scheduling or momentum-based optimizers). Our findings demonstrate that DOG can outperform EBMs with noisy dynamics and yields similar performance to "EBM w/ noise-free dynamics" on images. For a visualization of the generation optimization with intermediate samples, refer to Figure 7. The global structure is determined before local refinements are performed. For further insights, Figure 8 shows that both the corresponding generation loss and the norm of the gradients on the samples converge with minimal changes in the later time steps.

## B.4 FUNCTA LATENT SPACE

Moreover, for CelebA, we also perform an experiment in a latent space. We utilize the 256-dimensional functa representations for each image provided by (Dupont et al., 2022) and train a 4-layer MLP with a hidden size of 4096 and leaky ReLUs for $210,000$ steps to generate such representations. We employ a batch size of 128, Adam (Kingma & Ba, 2015) with $(\beta_1, \beta_2) = (0, 0.9)$, and

a learning rate of $1e - 5$. In GO, we use our default OneCycle (Smith & Topin, 2019) learning rate schedule with Adam and max_lr $= 0.1$. After training, decoding the generated representations yields face images. As shown in Figure 9, they appear to have higher quality than the ones in Figure 6(a), but they are evidently still worse than the current state of the art. Due to image blurriness caused by functa representations, Dupont et al. (2022) argue that an FID evaluation does not provide meaningful insights; hence, we do not report it. This experiment showcases that DOG is also applicable to a latent space domain.

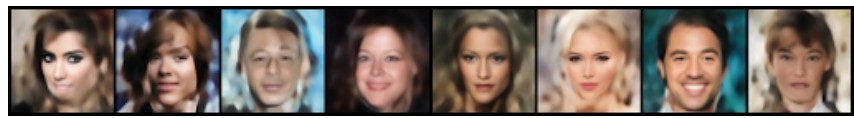

Figure 9: Generated samples for CelebA using DOG on functa.

## B.5 DISCUSSION

It is important to note the significant progress that has been made in generative modeling over the years, with numerous innovations required from early GANs (Goodfellow et al., 2014) to GigaGAN (Kang et al., 2023), and from early diffusion models (Sohl-Dickstein et al., 2015) to simple diffusion (Hoogeboom et al., 2023). Given the heavily studied field of image generation, we would be surprised to see near state-of-the-art results in the first paper exploring a generative modeling approach. We argue that extensive hyperparameter tuning, including substantial changes to the discriminator architecture and regularization techniques, may be required for DOG. For this demonstration, in Appendix B.2 we have only used StyleNAT's GAN-specific hyperparameters without modification and a slim version of the discriminator architecture and in Appendix B.3 we used very similar settings to EBMs, which typically also lag behind GANs in image generation.

Although some approaches developed for image GANs may be transferable to DOG, an exhaustive evaluation of them is beyond the scope of this paper. We believe that this should be addressed after making DOG faster by finding ways to reduce the number of GO steps $T$, as discussed in Section 6. Furthermore, it could be beneficial to employ regularization techniques, such as blurry gradients or progressive resolution, as utilized in (Wang & Torr, 2022) for generating images with pre-trained classification models. Additionally, methods that perform optimization in latent space, such as DragGAN (Pan et al., 2023), could prove advantageous.

The fact that existing GAN discriminator architectures (and also EBM energy estimator architectures) yield sub-par samples both in an EBM and in the DOG setting may also hint at the limited capacity of these architectures. Therefore, better DOG discriminator architectures that produce higher-quality samples may also be used to improve the performance of GANs.

Currently, DOG outperforms GANs on graphs, but not on images. We offer two potential reasons for this. First, the graph discriminators are much lighter than image discriminators, which means that we are not computationally limited by slow training steps. In fact, we trained DOG for the same number of steps, sometimes even using larger discriminators than the best graph GAN (SPECTRE) for QM9. However, for FFHQ, we used an order of magnitude fewer training steps and a much smaller discriminator than the best GAN, as we trained on a single Nvidia A40 for about a week. The second reason is that, given our assumptions, the GO process prefers to reach strict local maxima. Therefore, DOG may be better suited for generating discrete data (such as graphs where there is either an edge or none between two nodes) rather than continuous data such as images. There, a small amount of noise $\epsilon$ added to an image $\mathbf{x}$ would still be considered a valid sample, whereas DOG would only return $\mathbf{x} + \epsilon$ for a fixed subset of $\epsilon$, where $\mathbf{x} + \epsilon$ is a local maximum in the discriminator's score surface. To address this limitation, there are two potential remedies that can be explored in future work. One option is to use inference-time dropout in $D$ to create different local maxima for each GO. Another approach is to model pixel values as discrete rather than continuous.

Overall, DOG's strengths in graph generation showcase its potential. While most generative approaches are general in theory, some are better suited for specific domains than others. For example, GANs work well for images (Kang et al., 2023), but currently not for text (Alvarez-Melis et al., 2022).

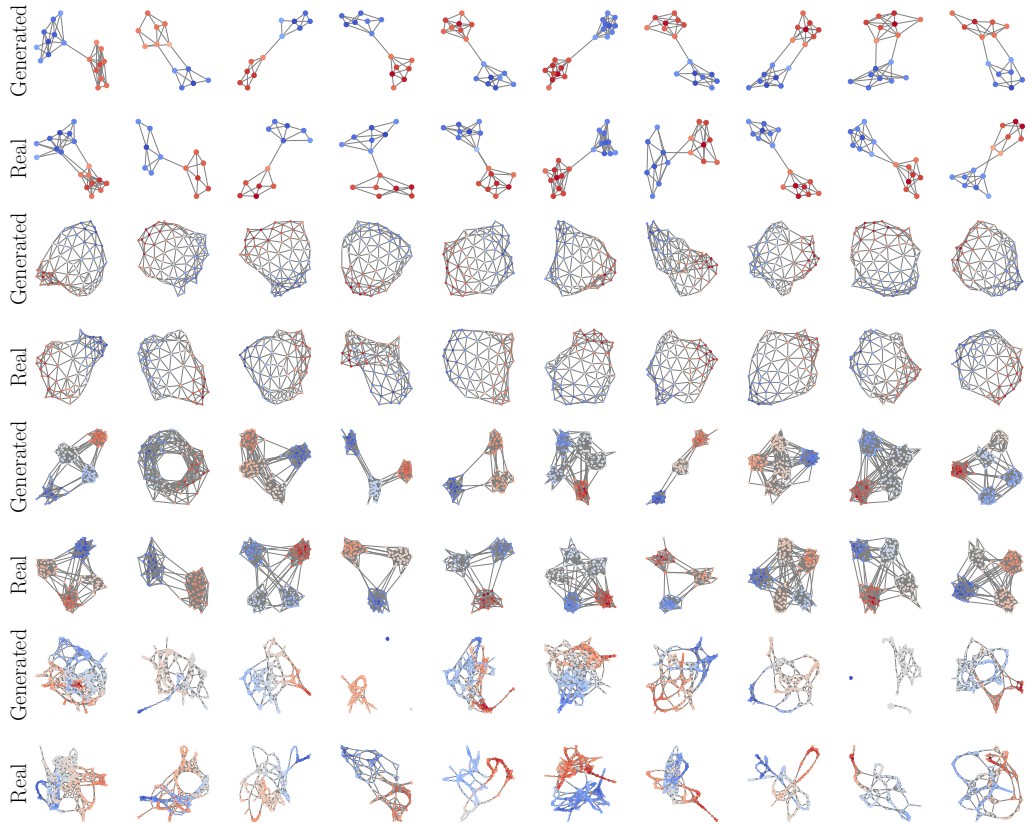

Figure 10: Uncurated set of DOG-generated and real samples for Community-small, Planar, SBM, and Proteins (from top to bottom). Many properties of the real data are covered by the generated samples. However, for example, for Proteins, disconnected components can occur: When a graph is approximately disconnected upon initialization, the message passing in the graph neural network discriminator does not allow for communication between the components. A simple solution would be to only keep the largest connected component. Alternatively, a thoughtful initialization of the adjacency matrix or constraining it during the generation optimization could help. For visualizations of generated graphs from GAN approaches, refer to (Martinkus et al., 2022).

## C   COMPUTATIONAL COST

To obtain an approximate comparison of the computational cost between DOG and GANs, we assume that the cost of each optimizer (i.e., weight/sample and momentum updates) is negligible. Additionally, we assume that the cost of a forward pass through $D$ is equal to the cost of a forward pass through a corresponding GAN generator model $G$, as $G$ and $D$ are often designed to be similar in size (Karras et al., 2020). Furthermore, we assume that a backward pass is twice as expensive as a forward pass.

For inference, a DOG GO requires $T$ forward and backward passes through $D$. Therefore, a single DOG generation costs $3T$ times as much as a generation with a $G$ that requires only a single forward pass through $G$. This brings the DOG inference cost closer to the cost of denoising diffusion models that perform dozens to thousands of forward passes (Ho et al., 2020).

When it comes to training, assuming a vanilla GAN with no augmentations or regularizations, we have a $G$ update step with one forward and backward pass through $G$ and $D$ (cost 6), and a $D$ update step with one forward pass through $G$ and two forward and backward passes through $D$, one for $\mathbf{x}^{\text{gen}}$ and one for $\mathbf{x}^{\text{real}}$ (cost 7). DOG does not perform a generator update step like GANs since there is no $G$. The discriminator update step includes a GO to obtain $\mathbf{x}^{\text{gen}}$ ($3T$ passes) and again two forward and backward passes through $D$ (cost 6). In total, we have 13 passes for a GAN training step versus

**Algorithm 2** Pseudocode for DOG in PyTorch-style. Here with $\mathcal{P} = \mathcal{N}(\mathbf{0}, \mathbf{I})$, Wasserstein losses, and SGD without learning rate scheduling.

```
1  d = Discriminator() # Randomly initialized model
2
3  def generation_optimization(shape): # Generate samples, don't update d
4      # Randomly initialize x_gen(=x_gen_0) as a parameter to be optimized
5      x_gen = Parameter(randn(shape))
6      x_gen_opt = SGD([x_gen], learning_rate_g)
7      for t in range(T):
8          x_gen_opt.zero_grad()
9          loss_g = -d(x_gen).mean() # Uses current parameters of d
10         loss_g.backward()
11         x_gen_opt.step() # Update x_gen = x_gen_t+1
12     return x_gen.detach() # Detach: stop gradient; d is not updated
13
14 # Discriminator training
15 d_opt = Adam(d.parameters(), learning_rate_d)
16 for x_real in dataloader:
17     x_gen = generation_optimization(x_real.shape)
18     d_opt.zero_grad()
19     loss_d = d(x_gen).mean() - d(x_real).mean()
20     loss_d.backward()
21     d_opt.step()
22
23 # Inference
24 x_gen = generation_optimization(x_real.shape)
```

$3T + 6$ passes for a DOG training step. For a typical $T = 100$, this theoretically makes DOG $\approx 23.5$ times slower than GANs.

In practice though, we could achieve higher accelerator utilization since no data loading bottlenecks can occur in the GO. Additionally, as quantified in the hyperparameter study (see Section 4.2), overall fewer training steps (and less time) might be needed for DOG.

## D  LIMITATIONS & SOCIETAL IMPACT

Like other generative approaches such as GANs or denoising diffusion approaches, DOG poses the risk of misuse or deception by malicious actors and may introduce biases from the training data into the generated samples. However, currently, the quality of the generated samples is only on par with diffusion approaches in the graph domain. Additionally, DOG training steps are computationally expensive, although fewer of them might be necessary compared to GANs.

