# OpenReview forum: "DOG: Discriminator-only Generation Beats GANs on Graphs"
_ICLR.cc/2024/Conference — Submitted to ICLR 2024_

### Official Review · Reviewer_2QS2 · 2023-10-23

**Soundness:** 2 fair
**Presentation:** 3 good
**Contribution:** 2 fair
**Rating:** 5
**Confidence:** 4

**Summary:**

The authors propose a generative model based on iterative gradient ascent of the discriminator. This generative model avoids the use of a generator during both training and sampling. During training, the discriminator learns from the data it generates itself. Experiments were conducted using both image and graph data. The model exhibits lower performance than standard GANs on image data, but it demonstrates superior performance over standard GANs on graph data.

**Strengths:**

* The paper is globally well written and structured, very easy to follow.

* The presented algorithm is clear and makes sense.

* The proposed generative model shows improved performance over standard GANs on graph data, although not SOTA.

**Weaknesses:**

* Overall, the paper has neither strong theoretical results, neither strong experimental results. Moreover, a very similar work is not discussed (see below), which lowers the contribution. Authors should at least discuss the differences with this method.

* The training process appears to lack a principled approach in the following aspect: there exists a mismatch between training and sampling times, which is typically an undesired characteristic. The discriminator generates samples by applying its gradient 'T' times, but the adversarial loss is only computed and back-propagated during the final step. This appears suboptimal since there is no guarantee that the gradients from the initial timesteps will be useful or provide appropriate guidance. Have there been instances of training runs failing to converge, or with catastrophic forgetting, and have you explored the possibility of learning from intermediate steps as well? The argument presented at the end of the section, suggesting that learning from intermediate steps would lead to memory issues, is not convincing. It is possible to choose to learn from intermediate timesteps with a 'stop-gradient' approach, which would not result in memory issues.

* The paper lacks theoretical justification for their generative model. The convergence analysis on a toy distribution is not sufficiently formalized and is a bit verbose. There is no explicit motivation or rationale provided for the use of their proposed generative model. Furthermore, while the argument regarding the min-max optimization is interesting, it does not clarify why their generative model would be superior in approximating distributions compared to a standard GAN; it primarily elaborates on how their approach might result in a better discriminator.

* Missing important references: 1: Mostly a recent similar work [1]. In this paper, the Discriminator Flow model is very similar to the one proposed by the authors. Discriminator Flow also consists in generating data with only a discriminator, and training the discriminator to distinguish between its own generated samples and training data. In this paper, the discriminator is trained on all intermediate timesteps, with timestep embedding.  2: Tanaka [2] which proposes to refine generated samples with discriminator's gradient. On image data, the updates are applied on the latent vector of the generator rather than directly on the image space.

[1] Franceschi, J. Y., Gartrell, M., Santos, L. D., Issenhuth, T., de Bézenac, E., Chen, M., & Rakotomamonjy, A. Unifying GANs and Score-Based Diffusion as Generative Particle Models. NeurIPS 2023.

[2] Tanaka, A. Discriminator optimal transport. NeurIPS 2019.

**Questions:**

* In diffusion models, embedding the time step seems to be a crucial component. Have you tried such strategy (i.e. embedding time step in the discriminator) in your model?

* It would be interesting to have some more analysis on the sampling procedure. For example, visualizations of the samples along the timesteps, at least on image data, or observing the evolution of the gradient's norm along the timesteps, to better understand the generative process.

---

> ### Author Response · Authors · 2023-11-14
>
> **W1/W4 References** Thank you for the references. Especially [1] is indeed complementary to our work, providing a formal connection between GANs and score-based diffusion. DOG is similar to a particle model. However, the DOG discriminator is not conditioned on timesteps, and the samples may be updated with momentum and learning rate scheduling instead of just the gradients. In contrast to DOG, their Discriminator Flow approach requires this conditioning for the discriminator and uses the intermediate samples for training, making DOG conceptually simpler. We also added [2] to its already referenced follow-up [3].
>
> **W2.1 Learning from Intermediate Timesteps** This is a great question! Earlier, we experimented with obtaining intermediate samples from a random (uniform) timestep of the generation instead of always taking the last one as a generated sample for computing the discriminator loss. This always quickly resulted in a failure mode with a very low discriminator loss (high generation loss) and poor samples that resembled pure noise. Since the samples from the initial timesteps are mostly noise, the discriminator loss encourages the discriminator to distinguish between noise and real samples, which is an easy task to learn. We hypothesize that such a discriminator that can detect noisy samples is unlikely to give meaningful gradients towards less noisy, more realistic samples. This indicates that naively using the intermediate samples in this setting actually hurts generation quality.
>
> In a variation to this experiment, we restricted the earliest time to be used for loss computation to be at least 50% of the total timesteps T. This yielded similar results to the default setting, albeit the training was less stable: the discriminator loss often sharply decreased and the sample quality deteriorated after some period of stable training with meaningful samples, indicating catastrophic forgetting. We observed similar behavior for default DOG training with smaller T. We think that most samples can reach a meaningful state in a few steps and are only minimally modified towards the end of the generation (see new figures 7 and 8). However, if enough samples that are used for loss calculation are still noisy, the above problem appears.
>
> But why does training only on the last samples nevertheless yield a discriminator that gives meaningful gradients to pure noise? While we do not have a definitive answer (p.9), potentially, these paths emerge in similar ways to the paths in classification models that adversarial reconstruction attacks take.
>
> **W2.2 Memory Issues** This argument was about keeping not only the intermediate samples but also their activations and gradients. Besides the then resulting memory issues, the discriminator could reduce its loss by giving zero gradients to noisy inputs. However, if we discard the activations/gradients, you are right, and we end up in a similar setting as described above.
>
> **W3.1 Theoretical Justification/Motivation** While we have not formalized this, the reason why we expected DOG to work is that we train the DOG discriminator similarly to the GAN setting: It learns to distinguish real and generated samples. Instead of taking the detour of first using the gradients from the discriminator to teach a generator to generate realistic (according to the discriminator) samples, DOG uses the gradients directly to generate realistic training samples via optimization.
>
> **W3.2 Advantages over Standard GANs/Motivation** While a generator ideally learns all the information about the data distribution contained in the discriminator, this might not be the case in practice: There may be certain limitations in the generator architectures. For instance, StyleGAN2 [4] and StyleGAN3 [5] use the same discriminator architecture, but StyleGAN3 achieves improved sample quality via a better generator architecture. One way to avoid these limitations (and generator architecture tuning) is to remove the generator altogether, leading to DOG. We show that DOG outperforms the best graph GAN, SPECTRE, that uses a complex setting with multiple generators. Furthermore, the iterative process allows for fixing previous inaccuracies in the generation, in contrast to the one-shot generation that GAN generators perform. Please also refer to our general comment.
>
> **Q1 Time Step Conditioning** We have not tried time step conditioning, as it is not directly applicable in the default setting: During training, the DOG discriminator is only trained with the real samples and the generated samples from the last timestep. It would be possible, though, as demonstrated in [1] and [6], if one meaningfully uses the intermediate samples for training.
>
> **Q2 Generation Visualization** This is a good idea; please see the updated appendix, in particular figures 7 and 8 (p. 19).

---

> > ### Author Response · Authors · 2023-11-14
> > **References**
> >
> > **References**
> >
> > [3] Abdul Fatir Ansari, Ming Liang Ang, and Harold Soh. Refining deep generative models via discriminator gradient flow. In International Conference on Learning Representations, 2020.
> >
> > [4] Tero Karras, Samuli Laine, Miika Aittala, Janne Hellsten, Jaakko Lehtinen, and Timo Aila. Analyzing and improving the image quality of stylegan. In Proceedings of the IEEE/CVF conference on computer vision and pattern recognition, pp. 8110–8119, 2020.
> >
> > [5] Tero Karras, Miika Aittala, Samuli Laine, Erik Härkönen, Janne Hellsten, Jaakko Lehtinen, and Timo Aila. Alias-free generative adversarial networks. Advances in Neural Information Processing Systems, 34:852–863, 2021.
> >
> > [6] Zhendong Wang, Huangjie Zheng, Pengcheng He, Weizhu Chen, and Mingyuan Zhou. Diffusion-gan: Training gans with diffusion. arXiv preprint arXiv:2206.02262, 2022.

---

> > > ### Comment · Reviewer_2QS2 · 2023-11-20
> > >
> > > Thank you for your answers. I appreciate that you revised the manuscript accordingly, adding the references and visualization.
> > > Let me note that I still disagree on some points, and think that there are areas of improvement:
> > >
> > > **Learning from intermediate timesteps** I am not convinced about your argument. You write "We hypothesize that such a discriminator that can detect noisy samples is unlikely to give meaningful gradients towards less noisy, more realistic samples." But how come a discriminator that would not learn on a noise distribution would give more useful gradients? Is it due to strong gradient penalty? Is it due to the discriminator learning meaningful gradients at the beginning of the training, when the generative optimization leads to noisy images? This could be more carefully studied and understood by the authors.
> > >
> > > **Memory issues**: Even taking into account activations/gradients, you could pre-generate in inference mode, and then generate the step i+1 in training mode.
> > >
> > > **More experiments on design choices**: especially, as mentionned above, on learning (or not) from intermediate timesteps, and on timestep embedding when learning from intermediate timesteps.
> > >
> > >
> > > In summary, I appreciate the innovative direction of utilizing the discriminator's gradients in the generation process. I concur with the authors' viewpoint that given its novelty in generative modeling, it's reasonable not to expect it to surpass SOTA, and such a factor should not be a basis for penalization. However, there is room for improvement in understanding the proposed method, and there seems to be a lack of important ablation studies.
> > >
> > > As it stands, I still perceive this paper as borderline, leaning towards a potential rejection due to the aforementioned concerns.

---

### Official Review · Reviewer_8BtC · 2023-10-23

**Soundness:** 3 good
**Presentation:** 2 fair
**Contribution:** 2 fair
**Rating:** 3
**Confidence:** 4

**Summary:**

This paper propose a discriminator-only generation (DOG) as a generative modeling approach. The DOG model generates samples through iterative gradient
descent on a discriminator’s input,  which is a commonly utilized technique in the field of adversarial attack research. In the graph domain, DOG demonstrates significant improvements overGANs using the same discriminator architectures.

**Strengths:**

This article introduces a novel approach by employing a single discriminator model for the generative process. This approach stands out because state-of-the-art generative adversarial networks typically rely on a pair of discriminator and generator models, making the training process time-consuming. Given the recent advancements in discriminator models such as CLIP, DINO, or SAM, utilizing these models as the discriminator to generate high-fidelity data has the potential to significantly advance the field of generative modeling.

**Weaknesses:**

The approach employed in the paper may appear to lack novelty. This is because the method of accumulating $\nabla_{x}D$ on the data point $x$ is a commonly utilized technique in the field of adversarial attack research. Moreover, in recent studies, certain generative models, such as those mentioned in references \cite{wang2022traditional} and \cite{dragan} have also adopted this method for tasks related to image generation and latent space editing.
Furthermore, the motivation of the article remains unclear and somewhat confusing. The author's underlying rationale for comparing the DOG method with the EBM approach is not explicitly communicated, making it challenging to discern the purpose of this comparison.
1. Wang, Guangrun, and Philip HS Torr. "Traditional Classification Neural Networks are Good Generators: They are Competitive with DDPMs and GANs." arXiv preprint arXiv:2211.14794 (2022).
2. Pan, Xingang, et al. "Drag your gan: Interactive point-based manipulation on the generative image manifold." ACM SIGGRAPH 2023 Conference Proceedings. 2023.

**Questions:**

1. I would suggest that the author provides additional clarification regarding the motivation behind comparing the EBM model with the DOG model. This clarification would be valuable in improving the comprehensibility of the research and elucidating the rationale for conducting this specific comparison.

2. In the "CONVERGENCE ANALYSIS" section, the reference to (Mescheder et al., 2018) is made in the context of using a dynamic system to establish the convergence of the gradient penalty. However, it is not entirely clear how the DOG model is related to this dynamic system. Additionally, there appear to be some mistakes in the convergence proof provided in the appendix.

3. In the "CONVERGENCE ANALYSIS" section, I remain perplexed by the choice to utilize the convergence of the Energy-Based Model (EBM) for comparison with the DOG model

4. There are several typos present in the article. For instance, in the "Related work" section, "EMB" is likely a typo and should be corrected to "EBM."


5. The presentation of the ablation study for hyper parameters in Table 6 is not accurate or correct

6. Are there any advantages to using the DOG model for graph data?I recommend that the author provide an advanced analysis.

---

> ### Author Response · Authors · 2023-11-14
>
> **W1 Novelty - Adversarial Attacks/Iterative Generation** As discussed on p. 8, DOG is a generative modeling approach aiming to generate novel samples, whereas adversarial reconstruction attacks aim to obtain training samples. While iterative generation using the gradient on a sample is indeed not novel (prominently EBMs use it), training a (graph) GAN discriminator from scratch to be suitable for this optimization without a generator is novel.
>
> **W2 References** Thank you for the references. In (Wang & Torr, 2022), iterative generation uses pretrained classifiers, not discriminators. Nevertheless, especially for the image domain, the regularization techniques used there, such as gradient blurring or progressive resolution, could be beneficial for DOG as well. In (Pan et al., 2023), DragGAN’s gradient-based optimization for editing operates in latent space where the optimization target is a specific manipulation, not explicitly realism as determined by a discriminator in DOG/GANs (or high density for EBM sampling). Instead, realism is achieved by the trained GAN generator. We added these references to the image domain discussion in Appendix B.5.
>
> **W3 Motivation** As described on p.1, graph GAN generators are complex, and designing them is tedious. We show that removing them and only using the discriminator (trained from scratch) can improve sample quality. The iterative generation is similar to EBMs and approaches that refine samples using a discriminator, and the training is similar to a GAN. Please also refer to our general comment.
>
> **W4/Q1/Q3 Reason for EBM Comparison** As foreshadowed on p.1 and elaborated on p.7, EBMs are closely related to DOG. Therefore, we point out a setting where DOG has advantages over EBMs. We made this more explicit now in Section 3.
>
> **Q2.1 Reference/Gradient Penalty** We only take inspiration from (Mescheder et al., 2018) regarding a small discriminator example. The training dynamics are not directly comparable because we only have a single model with a single trainable parameter, not two. Instead, we cover inner (generation) and outer (training) convergence. In this setting, DOG works without gradient penalty.
>
> **Q2.2 Proof Mistakes** There is no convergence proof in the appendix.
>
> **Q4 Typo** Thank you.
>
> **Q5 Hyperparameter Study** We now clarified that the individual experiments all follow the default setting with a single change, as indicated in the first column.
>
> **Q6 Advantages for DOG on Graphs** We show that DOG trains faster and achieves much better samples than GANs (Section 4.2). We discuss the simplified setting and potential reasons for faster training in Section 6 (p.8). For additional discussion, refer to Appendix B.5. Furthermore, DOG has an advantage over DiGress in that it requires fewer parameters and significantly fewer training epochs. For QM9, DiGress was trained for 1000 epochs using an expensive transformer model with 4.6 million parameters. In contrast, DOG achieves the same sample quality with only 30 epochs and 1.1 million parameters for the large discriminator configuration (and merely 145 thousand parameters for the default configuration). We added this analysis to the discussion. Please also refer to our general comment.

---

> > ### Comment · Reviewer_8BtC · 2023-11-18
> > **Review for the author**
> >
> > Thank you for responding to our review. I still concentrate on the novelty of the paper.  I agree the comments from the reviewer ssB3 that why the proposed method works better for the graph data? For the reason is that accumulating gradient on the data point x is not a new method which is commonly used in adversarial training\score-based model\EBM model. In that case, the special advantage using DOG on the graph datasets is the most important novelty of this paper. Can you describe this advantage from the theoretical perspective or from the intuitive opinion? Thank you.

---

> > > ### Author Response · Authors · 2023-11-19
> > > **DOG's advantages**
> > >
> > > Thank you for your reply!
> > >
> > > We provide a two-fold answer:
> > >
> > > ## First, we reiterate DOG’s advantages over EBMs and GANs, the two methods it is closest related to:
> > > In discrete settings described in Section 2, unlike DOG, EBMs would fail and also yield invalid samples. While EBMs may also use the gradients on the input of a model to update the input (e.g., when using Langevin Dynamics for sampling), they assume that the output of the model corresponds to a (positive) density from which we sample. However, for discrete data (such as the toy data in Section 2 or graphs), the density should be 0 for inputs that do not correspond to valid discrete inputs.
> > >
> > > Furthermore, in practice, EBMs often rely on imperfect sampling (p. 7), with a bias towards local minima in the energy surface (local maxima of the density). DOG avoids this discrepancy between theory and practice for EBMs. The generation optimization process is designed for sampling exactly from these local maxima (of the discriminator’s score surface), embracing the use of modern optimization techniques such as momentum.
> > >
> > > Compared to GANs, DOG has an advantage in that it does not rely on a generator. First, generator architectures are potentially restrictive and tedious to tune, where better sample quality often relies on better generator architectures, as discussed in our reply to reviewer 2QS2. Second, GAN generators produce samples in one go and do not benefit from the advantages of iterative generation such as DOG’s generation optimization. Third, the generated training samples that the discriminator receives may be more informative, leading to better discriminators with less training (p. 9): Instead of first having to teach the discriminator’s weaknesses to a generator, DOG directly uses the gradients from the discriminator to obtain samples that maximally exploit the discriminator’s weaknesses.
> > >
> > > ## Second, we reiterate the reasons for DOG’s success over graph GANs compared to image GANs (p. 20).
> > >
> > > First, image generator architecture tuning has received more effort than graph generator architecture tuning. The corresponding discriminator architectures are also tuned in conjunction with these generators, placing DOG at a disadvantage.
> > >
> > > Second, unlike graphs, images are not discrete, whereas DOG’s GO aims for local maxima in the discriminator’s surface. These maxima are potentially strict and therefore discrete.

---

### Official Review · Reviewer_5Xb1 · 2023-11-01

**Soundness:** 2 fair
**Presentation:** 2 fair
**Contribution:** 3 good
**Rating:** 6
**Confidence:** 3

**Summary:**

The authors present an innovative approach to training Generative Adversarial Networks (GANs) for graph tasks, utilizing only the discriminator in a process similar to diffusion models. They offer a novel iterative method that removes the traditional generator, proposing a training scheme that progressively refines sample generation through the interaction with the discriminator. A theoretical underpinning is provided with a convergence analysis demonstrated on simple 1-D grid data. Experiments across both synthetic datasets (such as 25-Gaussian, Community-small, and SBM) and real-world datasets (including QM9 and Proteins) demonstrate that this generator-free GAN approach yields results on par with existing methods like SPECTRE (GAN-based) and DiGress (Diffusion-based). Additionally, thorough ablation studies for factors, including  the number of channels, layers, optimization steps, and loss functions, providing some insight into the model's design choices.

**Strengths:**

1. The approach of training a discriminator without a generator is novel, particularly given its empirical effectiveness in graph tasks.


2. The authors present a method that achieves competitive performance with traditional GANs and diffusion-based models, offering a compelling alternative in the field. The authors' extensive research across graph and image generation tasks convincingly demonstrates that their method can produce quality samples without a generator, showcasing its efficacy.


3. The method is simple but useful.

**Weaknesses:**

1. While the authors provide a simple convergence analysis for their generator-free training method, the provided analysis falls short of explaining why removing the generator yields improved results for graph tasks, diminishing the persuasiveness of the approach.

2. Despite the proposed method's ability to operate without a generator, it incurs longer training times.

3. Table 6 includes ablation studies for several factors, dissecting the influence of steps, channels, layers, and optimizers separately, could yield a clearer, more convincing analysis.

**Questions:**

Please refer to the weaknesses part.

---

> ### Author Response · Authors · 2023-11-14
>
> **W1 Reason for Better Graph Generation** On p. 8, we discuss that the best graph generators (SPECTRE) are rather complex, requiring an elaborate approach where some generators and discriminators need to be trained before others. DOG removes all the generators and uses only a single discriminator. This greatly simplifies the setting and removes potential limitations from the generator architectures. Please also refer to our general comment.
>
> **W2 Longer Training Times** We actually observed the opposite for graphs, as mentioned in the abstract and on p. 7 (and discussed on p. 8): DOG trains faster than SPECTRE for QM9.
>
> **W3 Hyperparameter Study Clarity** We now clarified that the individual experiments all follow the default setting with a single change, as indicated in the first column.

---

> > ### Comment · Reviewer_5Xb1 · 2023-11-20
> > **Review for Authors**
> >
> > Thank you to the authors for their reply. After reviewing the comments, the rationale behind removing the generator from Graph remains ambiguous and unconvincing to me. Despite appreciating the bold effort to remove the generator for Graph learning, my concerns lead me to retain my original score.

---

### Official Review · Reviewer_SsB3 · 2023-11-02

**Soundness:** 2 fair
**Presentation:** 3 good
**Contribution:** 2 fair
**Rating:** 3
**Confidence:** 4

**Summary:**

This paper introduces a generative modeling approach for graph data.  It creates samples by iteratively optimizing the input to a discriminator, eliminating the need for a separate generator model. This simplification reduces the complexity of tuning generator architectures. The study shows that in the graph domain, where GANs have historically underperformed diffusion approaches in generating high-quality data, the proposed method outperforms GANs using the same discriminator architectures.

**Strengths:**

1) In this paper,  authors present a method to train GANs without generator. It successfully generate good images, which is convincing.

2) Paper is easy to follow.

3) Authors conduct effective experiments to support the proposed method.

**Weaknesses:**

I have a few questions, which is as following:

1) The proposed method is not new for me. Previous works already distill knowledge from the pretrained Discriminator into a new generator or noise (noise is update when optimizating). Thus I guess there are less contributions in this paper.

2) One import problem is why the proposed method works better for the graph data. I fail to find the reason why it works. I think it is important aspect that authors should explain.

3) How about the inference time? I guess I need to update a new noise again.

4) Although authors conducting a few experiments, the used datasets are small and little.

**Questions:**

My main concerns are two views: (1)  why it works better for the  graph data (2) it is not new without generator, since distilling knowledge from the pertained discriminator has been studied previously.

---

> ### Author Response · Authors · 2023-11-14
>
> **W1/Q2 Pretrained Discriminator - Novelty** Note that DOG does not employ a pretrained discriminator, where pretraining relies on another model, such as a generator. Instead, we train the DOG discriminator from scratch without any other model.
>
> **W2/Q1 Better Graph Performance** On p. 20, we reason that the advantages of DOG over GANs on graphs, compared to images, may stem from the lower computational load and the local maximum-seeking behavior of DOG that favors discrete samples (p. 3). Furthermore, on p. 8, we discuss that potential limitations from complex one-shot graph generator models, such as in SPECTRE, are avoided in the multi-step DOG, which can correct inaccuracies from previous steps. Additionally, training progresses faster (p. 7), possibly because DOG's generation optimization provides more informative training samples to the discriminator (p. 8f). Please also refer to our general comment.
>
> **W3 Inference Time** As described on p. 2, in lines 1-7 of Algorithm 1, and in Appendix C, the inference uses new noise and the same iterative generation that is used in training.
>
> **W4 Small Datasets** We respectfully disagree. We use the same common graph generation benchmarks as our main baseline, SPECTRE. Also, for images, CelebA, for instance, is a common generation benchmark, which was also used in the EBM baseline we compare against.

---

### Author Response · Authors · 2023-11-14
**Revision**

Dear Reviewers,

We would like to thank you for your insightful feedback.

We updated the manuscript with the following changes:

- Added a paragraph about the high cost of DiGress (p. 8)
- Added figures 7 and 8 showing image samples during generation and the generation loss as well as the gradient norm during generation (p. 19)
- Improved clarity for the hyperparameter table
- Added references
- Fixed typos
- To accommodate the page limit, we moved the “Limitations & Societal Impact” section to the appendix.

We further reiterate **DOG’s advantages**: DOG avoids using complex and tediously tuned GAN generators, retaining only a discriminator and thereby avoiding limitations that some generators may have. The DOG discriminator is trained from scratch, similar to a GAN setting, but it utilizes samples generated directly from the information stored in the discriminator. This eliminates the need to train a generator first, thereby avoiding potential information loss. This approach may result in more informative training samples, as evidenced by faster training progress compared to the best graph GAN and a recent diffusion-based approach. The samples are iteratively generated using gradients from the discriminator, akin to EBMs and methods that refine samples through a discriminator. However, DOG is more flexible, allowing for modern optimization techniques such as momentum in the generation. DOG also exhibits advantages in discrete settings over EBMs and does not depend on other models or time-step conditioning, making it conceptually simple. Additionally, the iterative process enables the correction of previous inaccuracies in the generation, in contrast to the one-shot generation performed by GAN generators.

Please let us know about any unresolved points.

Sincerely,

The Authors

---

### Meta-Review · Area_Chair_MsLD · 2023-12-14

**Metareview:**

This paper proposes a generative model for graphs based on direct backpropogation of a discriminator loss on the input space. The authors project this work as an analog of GANs, without any generator. While the empirical results show promise, the reviewers had several concerns about the the novelty of the work when perceived broadly in the field of generative modeling, as similar approaches have been explored for both GANs and EBMs for other data modalities. While graph generation in itself is valuable, the other major concern was that the design choices in the execution of the algorithm (such as ignoring the intermediate losses for backprop) were poorly justified. Many of these concerns persisted even after the discussion period. A more thorough investigation into the merits of the approach and clearer distinctions with prior approaches (perhaps even as ablation baselines) can improve a future version of this paper.

**Justification For Why Not Higher Score:**

Many common concerns relating to novelty, theoretical and empirical rigor.

**Justification For Why Not Lower Score:**

N/A

---

### Decision · Program_Chairs · 2024-01-16

Reject